# Availability and factors influencing community level handwashing facility in Ethiopia: Implication for prevention of infectious diseases

Daniel Bogale Odo[1]*, Alemayehu Gonie Mekonnen[2]

1 Department of Public Health, College of Health Sciences, Arsi University, Asela, Oromia Regional State, Ethiopia, 2 Department of Nursing, College of Health Science, Debre Berhan University, Debre Berhan, Amhara Regional State, Ethiopia

These authors contributed equally to this work.

* dbogale386@gmail.com

**Data Availability Statement:** The data used in this study are available from The DHS Program database, which can be accessed with registration

## Abstract

### Background

Handwashing is one of the most effective ways to prevent transmission of infectious diseases. A substantial body of research has examined the status and determinants of handwashing facilities in healthcare settings and schools. However, its status at home in the community, especially in developing countries, remains unclear. This study aimed to examine the availability and factors influencing basic handwashing facilities at households in Ethiopia.

### Method

We analysed the 2016 Ethiopian Demographic and Health Survey (EDHS) data. EDHS employed a two-stage stratified cluster sampling technique. Data were collected from the lowest administrative unit (kebele). A multivariable logistic regression model that allowed cluster-level random effects was employed to examine factors that affect the availability of basic handwashing facilities (water plus soap) at households. Estimates from the regression model are reported as odds ratios (ORs) with standard errors clustered at the DHS cluster level to account for a sampling methodology.

### Results

In our sample, only 1292 (8% [95% CI, 7.6%–8.4%]) of the households had basic handwashing facilities. Compared with head of household who had no formal education, the odds of having basic handwashing facilities was higher among head of household who completed secondary level of education (adjusted odds ratio [AOR] = 1.83; 95% CI: 1.35–2.49) and higher level of education (AOR = 2.35; 95% CI: 1.63–3.39). Odds of having basic handwashing facilities was increased with having radio (AOR = 1.32; 95% CI: 1.10–1.63) and television (AOR = 1.49; 95% CI: 1.10–2.02) at home. Households that had improved latrine were two times more likely to have basic handwashing facilities (AOR = 2.09; 95% CI: 1.56–

in https://www.dhsprogram.com/data/dataset_admin/login_main.cfm.

**Funding:** The authors received no specific funding for this work.

**Competing interests:** The authors have declared that no competing interests exist.

2.80). Being at higher household wealth quintiles was associated with increased odds of having basic handwashing facilities.

## Conclusion

Very low basic handwashing facilities was demonstrated by this study, whereas, awareness and socio-economic related factors were identified as a determinants for its availability in the household. Greater efforts are needed to increase the coverage of community-level handwashing facilities.

## Introduction

Handwashing with soap is one of the most effective and inexpensive means of reducing the spread of infectious diseases [1]. Presence of designated handwashing facilities in the household promote handwashing practice at all critical times [2]. Dedication of October 15 as a global handwashing day, which has been declared by UNICEF since 2008, is signalling the worldwide importance of handwashing. However, according to the joint report from the World Health Organization (WHO) and UNICEF, two out of five people in the world do not have basic handwashing facilities (water plus detergents), and an overwhelming number of these people are from the least developing countries [3]. As a result, many of the least developed countries bear the highest brunt of disabilities and deaths that are related to many communicable diseases which could be averted by effective sanitation and hygiene program. Gastrointestinal and respiratory infections are disproportionately affecting such countries [4, 5], and pandemic diseases such as coronavirus (COVID-19), which can be prevented by effective hand hygiene, are a threat to people who are living in resource-limited countries including Ethiopia.

According to WHO, 88% of diarrhoea cases worldwide are attributed to unsafe water, inadequate sanitation or insufficient hygiene, and these cases result in 1.5 million deaths each year [6]. Washing hands with water and soap at all critical times has the potential to save millions of lives, but close to one-third of the people in the least developed nations lack basic handwashing facilities at home [3]. Three countries including Ethiopia, Nigeria and DR Congo account for one-third of the sub-Saharan Africa population that do not have basic handwashing facilities at home [3]. Systematic reviews and meta-analyses on the importance of handwashing indicated that it has substantial contribution in reducing gastrointestinal and respiratory illness [1, 7, 8]. A study conducted among school children in Egypt showed that effective handwashing reduced school absenteeism because of acute respiratory infections, diarrhoea and conjunctivitis by 40%, 30% and 67%, respectively [9].

Existing research and government reports showed that Ethiopia is facing an overwhelming number of disabilities and deaths of under-five children that are attributed to gastrointestinal and respiratory diseases [10, 11]. Inadequate and inefficient infrastructures such as water, sanitation, housing and personal hygiene facilities are responsible for such unacceptable burden of diseases and deaths [10]. Additionally, Ethiopia is facing frequent public health emergencies such as drought and social conflicts that have been leading to displacement and recurrent outbreaks of infectious diseases. Despite significant and continuous efforts by the government to strengthening healthcare system, Ethiopia has very few healthcare facilities that do not commensurate with the fast-growing number of population in the country [12]. For instance, currently the country has less than 500 functional ventilators, of which only 50 dedicated to

treating COVID-19 patients [13]. These public health emergencies together with infrastructure related challenges mentioned therein make the country highly susceptible to communicable and pandemic diseases-related adverse health and economic effects.

Factors found to be influencing availability of basic handwashing facilities and consistent behaviour towards handwashing practice in the health care settings [14] and schools [8] have been explored in several studies. Existing community-based research to date has tended to focus on evaluation of sanitation and hygiene-related interventions rather than nationwide coverage and the associated factors [15, 16]. As a result, despite the importance of community-level handwashing facility, there remains a paucity of evidence on the coverage and the influencing factors. Very few studies identified inaccessibility of water, absence of sanitation facility (latrine) [17], lower level of household education, limited sanitation related information and large number of family size [10, 18] as a determinant for the availability of basic handwashing facilities in the community.

Also, some studies have been conducted on hand hygiene-related awareness in Ethiopia [10, 19], but there has been no such a nationwide analysis on the availability and factors affecting handwashing facility in the country. The objective of this analysis was to describe availability of basic handwashing facilities and examine factors affecting its presence at households in the community in Ethiopia. Results from this study will assist the promotion of installation of basic handwashing facilities and practice, which could in turn play a substantial role in reducing the spread of infectious diseases and support future research.

## Methods

### Source of data

Data on basic handwashing facilities obtained from the 2016 Ethiopian Demographic and Health Survey (EDHS), a nationally representative household survey that is conducted to estimate population, health, nutrition, and environment related indictors at regional and national levels. These data were collected from the lowest administrative unit (kebele). The 2016 EDHS followed a two-stage stratified cluster sampling, in which clusters were selected from a list of enumeration areas (EAs) formed for the 2007 population census (primary sampling unit), and then households were randomly selected in each of the selected clusters (secondary sampling unit). Details on sampling procedure used by EDHS is presented elsewhere [11]. A total of 16650 households (5,232 urban and 11,418 rural) from 645 sampling points (clusters) were included in the analysis. The survey was conducted between 18th of January 2016 to 27th of June 2016.

### Dependent variable

The outcome of interest for this study was basic handwashing facility. The EDHS enumerators collect handwashing-related information such as place where household members wash their hands through face-to-face interview and confirm presence of water, soap, and other detergent (ash, mud, or sand) on premises by observation. Based on this information, households that had a fixed place where household members wash their hands and those who had water plus soap on premises (confirmed by observation) at the time of the interview, were considered as having basic handwashing facilities. This approach (confirmation of availability of handwashing facility and presence of water plus soap on the premises by observation) is used to drive proportion of basic hand hygiene facility globally [20], and is the preferred approach next to randomized controlled design to study handwashing behaviour [21]. Details on definitions used for all handwashing related variables is presented (Table 1).

**Table 1. Description and classification of water, hygiene, and sanitation (WASH) related variables included in the analysis (based in WHO standard), EDHS 2016.**

| Variables | Descriptions |
|---|---|
| Basic handwashing facility | Handwashing facility present and availability of soap and water on premises observed during interview |
| Limited handwashing facility | Handwashing facility present but neither water nor detergent observed on premises |
| Water source | **Improved**: When water source is: piped water (piped into dwelling, piped to yard/plot, piped to neighbour), public tap/standpipe, tube well, or borehole |
| | **Unimproved**: When water source is: dug well (open/protected), protected well, unprotected well, surface (spring, river, dam, lake, ponds, stream, canal or irrigation channel), protected spring, unprotected spring, rainwater, or tanker truck |
| Time to get water | Water available on premises plus accessed in less than 30 minutes in round trip vs. requires greater than 30 minutes to access |
| Sanitation facility | **Improved**: flush toilet (flush to piped sewer system, flush to septic tank, flush to pit latrine), ventilated improved pit latrine (VIP), pit latrine with slab or composting toilet |
| | **Unimproved**: flush to somewhere else but do not know where, pit latrine without slab/ open pit, no facility, no facility/bush/field, bucket toilet and hanging toilet/latrine |
| Housing condition (floor) | **Standard**: finished, parquet or polished wood, vinyl or asphalt strips, ceramic tiles, cement, or carpet |
| | **Sub-standard**: natural, earth/sand, dung, rudimentary, wood planks and palm/bamboo |

## Independent variables

To assess individual and household characteristics that affect the availability of community-level basic handwashing facility, the following variables were extracted for each household: water source (improved vs. unimproved), water accessibility (time to get water), status of sanitation facility (improved vs. unimproved) [10, 17], floor material (standard vs. substandard), media exposure (having radio and/or television), educational status of head of household, sex of head of household, age of head of household, household wealth index and place of residence [10, 22]. Below table gives details on definition and coding of some of these variables (Table 1).

## Data analysis

In order to correct the EDHS sampling issues such as over/under sampling (disproportionate sampling), and non-responses, all descriptive statistics were produced after sample weights were applied to the data, based on recommendation by the EDHS [11]. Multivariable logistic regression model was used to assess factors affecting availability of community-level basic handwashing facility. Because of the hierarchical nature of the data in which households were sampled within clusters, estimates from our regression were reported as odds ratios (ORs) with standard errors clustered at the DHS cluster level. And cluster level random effects that allowed correlations between outcomes for households within a DHS cluster was accounted. A p-value of <0.05 was used to declare statistical significance.

All analyses were conducted using STATA software (version STATA/SE 16; StataCorp LP, College Station, Texas, USA).

## Ethics statement

The survey was approved by the National Research Ethics Review Committee of Ethiopia and ORC Macro Institutional. We got permission from ICF-DHS program on August 27, 2019, and there were no names of individuals or household addresses in the data file we received. Geographic information system (GIS) collected only for the sampling points (enumeration

areas), not for individual households, and measured coordinates are further displaced within a large geographic areas so that specific enumeration area cannot be identified. We accessed the dataset through 'https://www.dhsprogram.com'.

## Results

The weighted descriptive result showed that only 8% (95% CI, 7.6%– 8.4%) of households had basic handwashing facilities. Of those households reported as having handwashing facility (at fixed or mobile place), presence of water on the premises were confirmed by observation in about a third (32.44%) of households. As well, presence of soap on the premises confirmed by observation in about one-fifth (19.75%) of households, and 53.04% of households identified as having limited handwashing facility (handwashing place available but neither water nor detergent observed on premises) (Table 2).

Hot spot areas of handwashing facilities (clusters where relatively large number of households that had basic handwashing facilities recorded) were detected in the capital city, Addis Ababa, and in a few places of Oromia region surrounding Addis Ababa (99% confidence hot spot areas). Also, hot spot areas of handwashing facilities were observed in a few clusters of Southern Nation, Nationalities People's (SNNP) and Amhara regions. Cold spot areas of handwashing facilities (clusters where relatively small number of households that had basic handwashing facilities recorded) were observed in Amhara, Gambela and SNNP regions (99% confidence cold spot areas). As indicated by black dots, majority of the clusters had very small number of facilities, and there were also clusters with no handwashing facility at all (white dots) (Fig 1).

Table 3 presents proportions of individual and household characteristics and their crude association with availability of basic handwashing facilities at community-level in Ethiopia. One-fourth of households were headed by females, and of the total sample (households headed by male and female), 54.81% of household heads had no formal education. More than one-fourth (28.21%) and 13.76% of households had radio and television, respectively. About 47% of households had improved water source, and more than half (51.76%) of the interviewed households reported that they had spent less than 30 minutes to get water. Only 14.84%

**Table 2. Weighted descriptive statistics of household-level handwashing facilities in Ethiopia, EDHS 2016.**

| Variables | | households | Percent |
|---|---|---|---|
| Fixed place for handwashing | Yes | 592 | 3.58 |
| | No | 15920 | 96.42 |
| Mobile place for handwashing | Yes | 9273 | 43.84 |
| | No | 7239 | 43.84 |
| Fixed or mobile place for handwashing | Yes | 9865 | 59.74 |
| | No | 6647 | 40.26 |
| Place observed for handwashing with water (n = 9865) | Yes | 3200 | 32.44 |
| | No | 6665 | 67.56 |
| Place observed for handwashing with soap (n = 9865) | Yes | 1948 | 19.75 |
| | No | 7917 | 80.25 |
| Place observed for handwashing with cleansing agent other than soap (n = 9865) | Yes | 93 | 0.94 |
| | No | 9772 | 99.06 |
| Basic handwashing facility | Yes | 1292 | 7.99 |
| | No | 14873 | 92.01 |
| Limited handwashing facility | Yes | 8573 | 53.04 |
| | No | 7592 | 46.96 |

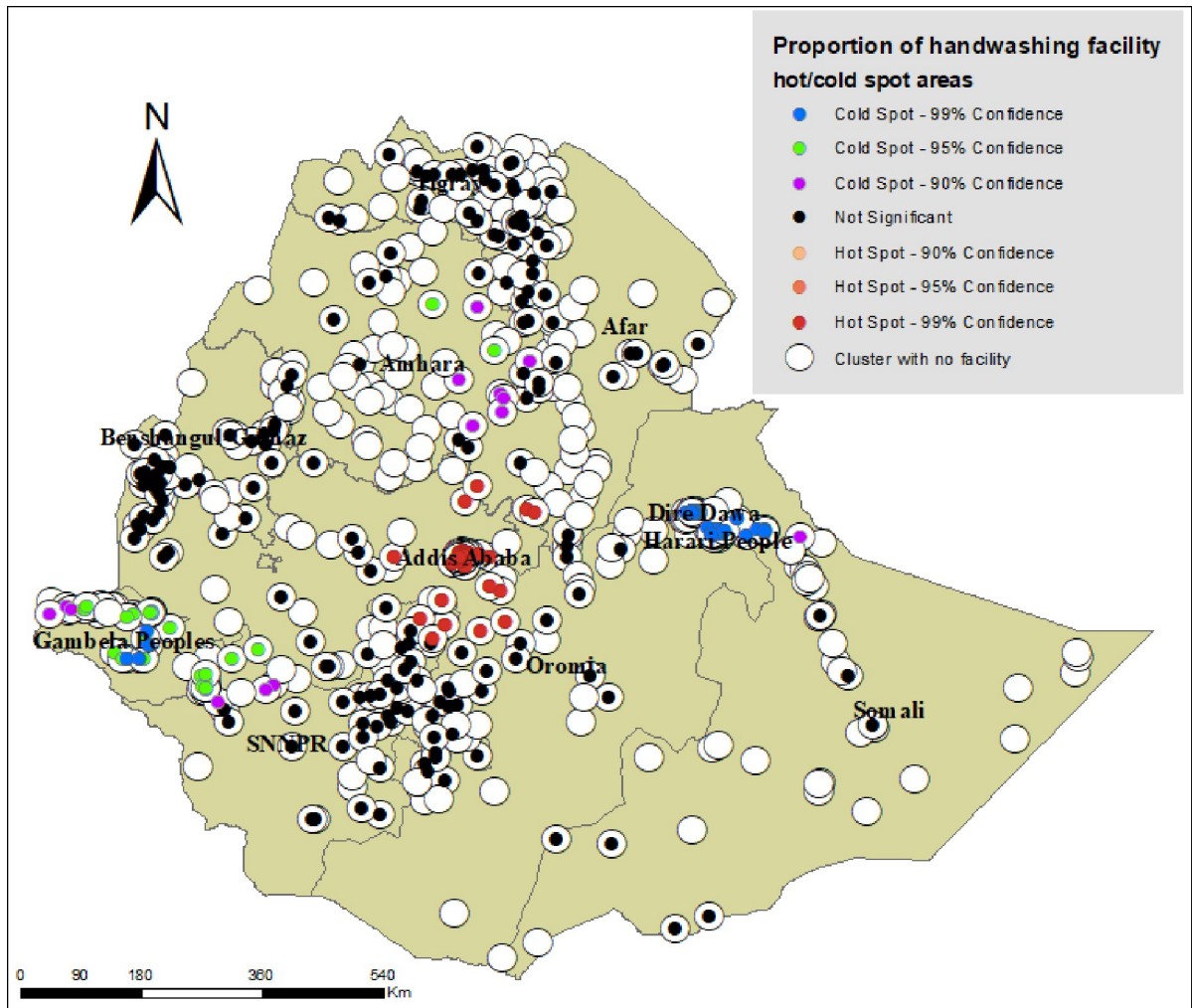

**Fig 1. Map portraying hot spot and cold spot areas of basic handwashing facilities in Ethiopia, based on the 2016 EDHS.**

households had improved sanitation facility. Almost all factors included in the analysis were associated with availability of basic handwashing facilities (Table 3).

## Multivariable logistic regression results

Five of twelve variables included in multivariable logistic regression model were associated with availability of basic handwashing facility. As indicated in Table 4, the likelihood of having basic handwashing facilities at home increased with educational status of head of the household. Specifically, compared to head of household who had no formal education, odds of having basic handwashing facilities was 1.83 times higher for head of household who had attended secondary education (AOR = 1.83; 95% CI: 1.35–2.49), and 2.35 times higher for head of household who had attended higher education (AOR = 2.35; 95% CI: 1.63–3.39).

Presence of radio and television in households had a significant and positive effect on availability of basic handwashing facilities. Odds of having basic handwashing facilities was increased by 32% in those households which had radio (AOR = 1.32; 95% CI: 1.10–1.63), and by 49% in those households which had television (AOR = 1.49; 95% CI: 1.10–2.02) (Table 3).

**Table 3. Weighted proportion of household and individual characteristics affecting availability of basic handwashing facilities at home in Ethiopia, EDHS 2016.**

| Variables | | Have basic handwashing facility | | P value |
|---|---|---|---|---|
| | | No (%) | Yes (%) | |
| Age of head of household | <20 years | 234 (95.12) | 12 (4.88) | <0.001 |
| | 20–29 years | 2650 (88.48) | 345 (11.52) | |
| | 30–39 years | 3675 (90.65) | 379 (9.35) | |
| | 40–49 years | 2749 (90.04) | 304 (9.96) | |
| | > = 50 years | 5265 (91.12) | 513 (8.88) | |
| Sex of head of household | Male | 10027 (90.54) | 1048 (9.46) | 0.285 |
| | Female | 4546 (90.00) | 505 (10.00) | |
| Educational status of head of household | No formal education | 7978 (95.59) | 368 (4.41) | <0.001 |
| | Primary | 4104 (90.66) | 423 (9.34) | |
| | Secondary | 1332 (80.97) | 313 (19.03) | |
| | Higher | 1108 (71.44) | 443 (28.56) | |
| Family size | < = 3 | 5557 (88.87) | 696 (11.13) | <0.001 |
| | 4–7 | 7419 (90.93) | 740 (9.07) | |
| | >7 | 1597 (93.17) | 117 (6.83) | |
| Has radio | No | 10546 (93.40) | 745 (6.60) | <0.001 |
| | Yes | 4027 (83.29) | 808 (16.71) | |
| Has television | No | 11769 (95.09) | 608 (4.91) | <0.001 |
| | Yes | 2804 (74.79) | 945 (25.21) | |
| Water source | Unimproved | 6968 (95.67) | 315 (4.33) | <0.001 |
| | Improved | 7605 (86.00) | 1238 (14.00) | |
| Time to get water | >30 minutes | 7914 (87.07) | 1175 (12.93) | <0.001 |
| | < = 30 minutes | 6659 (94.63) | 378 (5.37) | |
| Sanitation facility | Unimproved | 11369 (94.82) | 621 (5.18) | <0.001 |
| | Improved | 3204 (77.47) | 932 (22.53) | |
| Housing condition (floor) | Sub-standard | 10975 (95.39) | 531 (4.61) | <0.001 |
| | Standard | 3598 (77.88) | 1022 (22.12) | |
| Household wealth index | Poorest | 4353 (98.37) | 72 (1.63) | <0.001 |
| | Poorer | 2220 (96.65) | 77 (3.35) | |
| | Middle | 1921 (95.24) | 96 (4.76) | |
| | Richer | 1827 (92.32) | 152 (7.68) | |
| | Richest | 4252 (78.62) | 1156 (21.38) | |
| Place of residence | Urban | 4002 (78.67) | 1085 (21.33) | <0.001 |
| | Rural | 10571 (95.76) | 468 (4.24) | |

We incorporated source of water (improved vs. unimproved), accessibility of water (in terms of time require to get), and sanitation facility (improved vs. unimproved) as a determinant for availability of basic handwashing facilities. Accordingly, households that had improved sanitation facility (latrine) were 2 times more likely to have basic handwashing facilities (AOR = 2.09; 95% CI: 1.56–2.80) compared to counterparts with sub-standard sanitation facility. Source of water and time require to get water were associated with availability of basic handwashing facility only at bivariate analysis level (Table 4).

Household wealth index, which was computed based on selected assets owned by households, identified as a significant factor to have basic handwashing facilities at home. As it can be seen in Table 4, the odds of having basic handwashing facilities was about 5 times higher in households from the highest (richest) wealth index (AOR = 4.98; 95% CI:2.66–9.33) compared to households in the lowest wealth index (poorest). Also, compared to households in the

**Table 4. Multivariable logistic regression of factors affecting availability of basic handwashing facilities in Ethiopia, EDHS 2016.**

| Variables | AOR (95% CI) | P value |
|---|---|---|
| Age of head of household | 1.00 (0.99, 1.01) | 0.519 |
| Sex of head of household (reference: Male) | | |
| Female | 1.11 (0.87, 1.40) | 0.403 |
| Educational status of head of household (reference: No formal education) | | |
| Primary | 1.24 (0.96, 1.61) | 0.103 |
| Secondary | **1.83 (1.35, 2.49)** | **<0.001** |
| Higher | **2.35 (1.63, 3.39)** | **<0.001** |
| Family size | 1.02 (0.98, 1.06) | 0.282 |
| Has radio (reference: No) | | |
| Yes | **1.32 (1.10, 1.63)** | **0.010** |
| Has television (reference: No) | | |
| Yes | **1.49 (1.10, 2.02)** | **0.012** |
| Water source (reference: Unimproved) | | |
| Improved | 0.96 (0.69, 1.33) | 0.797 |
| Time to get water (reference: greater than 30 minutes) | | |
| Less or equal to 30 minutes | 1.08 (0.84, 1.39) | 0.563 |
| Sanitation facility (reference: Unimproved) | | |
| Improved | **2.08 (1.56, 2.77)** | **<0.001** |
| Housing condition (floor) (reference: Sub-standard) | | |
| Standard | 1.26 (0.99, 1.59) | 0.056 |
| Household wealth index (reference: Poorest) | | |
| Poorer | **2.51 (1.44, 4.40)** | **0.001** |
| Middle | **2.74 (1.61, 4.69)** | **<0.001** |
| Richer | **3.95 (2.32, 6.72)** | **<0.001** |
| Richest | **4.99 (2.66, 9.33)** | **<0.001** |
| Place of residence (reference: Urban) | | |
| Rural | 0.71 (0.44, 1.12) | 0.142 |

AOR–adjusted odd ratio, CI–confidence interval.

reference category, those in the second, third and fourth wealth quintiles were more likely to have basic handwashing facilities (Table 4).

## Discussion

This study set out with the aim of assessing the availability and factors influencing community-level handwashing facilities in Ethiopia. Overall, our finding revealed that more than 90% of households had no basic handwashing facilities. This study also highlighted factors that were associated with the availability of basic handwashing facilities at the community-level. To improve the health status of its citizen in an equitable manner, Ethiopian government has implemented a successive Health Sector Development Plans (HSDPs) with the key concept of ensuring community ownership and enabling them to produce their own health [23]. One of the globally noted components of Ethiopian HSDPs is health extension program (HEP), which has a total of 16 packages, of which 7 are environmental health related [24]. Despite such tremendous efforts made by the government, the current study found that fewer than 10% of households in Ethiopia had basic handwashing facilities. This implied that the country is far behind the Sustainable Development Goals (SDGs) ambition to achieving access to sanitation

and hygiene services by the year 2030 [25], and continued to suffer from infectious diseases which could potentially be prevented by washing hands with water and soap [5].

As to the factors determining availability of basic handwashing facilities at households in the community, basic handwashing facilities observed in households where heads had completed secondary and above educational level compared to counterparts who had no formal education. This finding is supported by previous literature [26]. This is plausible finding in that education is a foundation for weighing advantage and disadvantage of washing hands at critical times and develop behaviour towards the practice. However, in terms of consistent handwashing practice, education may not be sufficient to develop consistent handwashing behaviour [27] because this behaviour requires continuous and coordinated social and organizational triggers to be sustained [28].

The current study showed that having radio and television at home had a significant effect on the presence of basic handwashing facilities in the community. This is an interesting result in that radio and television are effective and efficient means to undertake coordinated nationwide awareness creation campaigns. Recently the International Committee of the Red Cross (ICRC) has funded the production of television and radio spots aimed at raising public awareness on COVID-19 in Ethiopia [29], which is signalling the indispensable role of these media to heightened community awareness. This media-based water, sanitation and hygiene (WASH) related campaigns will ensure reachability of messages at a broader audience because: (1) households or at least their neighbours could already have one of these means (no extra cost required for establishment), (2) easy to reach those households in rural and remote places, and (3) awareness creation program which is developed at the centre (e.g., by the ministry of health) can be broadcasted swiftly in an organized manner. In fact, existing evidence supported the role of media exposure on enhancing handwashing practice at the community level [30], and interventions on handwashing promotion have identified effective in reducing morbidity and mortality from infectious diseases [7]. Moreover, promotion of significant community coverage for sanitation appears to be essential to achieve multiple targets of the SDGs.

This analysis revealed that there was a positive and significant association between presence of improved sanitation facility (latrine) in households and availability of basic handwashing facilities. One potential justification for this is that presence of constructed latrine in households enhances long-term utilization of the system, which could in turn facilitate placement of permanent handwashing facilities on premises [31]. This specific finding suggests that efforts that undertaken to expand latrine coverage worldwide would not only help to meet latrine coverage-driven targets, but also an increase of basic handwashing facilities in the community. This result is consistent with that from other studies [17, 32]. While it showed association in the previous study [17], we found no statistically significant difference in households with improved water sources and in those households with easy access to water on the availability of basic handwashing facilities. Our finding of no effect is interesting in that households could manage to allocate water for handwashing regardless of its source and accessibility.

Household wealth index, a composite index which was computed from household facilities and possessions such as water source, sanitation facility, type of flooring, electricity, number of living rooms, radio, television, phone (landline or cell phone), motorcycle and car, was associated with having basic handwashing facilities. Compared to households in the first wealth quintile, those in the second, third, fourth and fifth quintiles were more likely to have basic handwashing facilities. This result may be explained by the fact that this household owned assets-based wealth index helps to show the status of households in terms of income and livelihood standards, which could in turn determine the household's ability to install basic handwashing facilities. The current result is in line with existing findings where a positive correlation between economic status and handwashing facilities at the household [10, 26] and

the country [2] level were reported. This finding has important implications for developing targeted sanitation- and hygiene-related subsidies in the community.

The current study is brought the status of community-level handwashing facilities in Ethiopia based on good quality information source (presence of water and soap on premises confirmed through observation), which is the preferred method next to randomized controlled design [21]. Obviously, measuring availability and utilization of handwashing facility through a questionnaire-based interview is often liable for a socially desirable answers [20]. However, our analysis overcame this shortcoming by augmenting the face-to-face interview data with observation. The finding has a substantial contribution for the country to plan targeted interventions on availability and utilization of handwashing facilities, which could in turn have a substantial benefit to halt the spread of infectious diseases like COVID-19 in the community. Finding from the current study is also a reflective of the status of WASH in many developing countries so that can be used to build community awareness on importance of handwashing practice. A large sample size increases credibility of our conclusion. In addition to assessing status of handwashing facilities, the current analysis helps to glimpse status of other WASH related components such as water and sanitation facilities in Ethiopia that require equal attention with the topic we have investigated.

Our study is not without limitation so that the following points need caution when results are interpreted. Even though DHS is the best, often only, available source of information on a wide range of public health issues in Ethiopia, it is a cross-sectional by nature–it reflects handwashing status that exists only at the time of the survey. DHS is not specific to WASH, therefore, all the variables that need to be considered with handwashing (e.g., social, cultural, behavioural and WASH-related policies) were not collected. As a result, despite our findings are not far from the existing evidence, there could be other factors confounding or mediating the current associations. Availability could not be necessarily signalling practice, which is more of behavioural that requires many and frequent triggering factors, so that needs follow-up studies.

## Conclusion

Less than ten percent of households had basic handwashing facilities in Ethiopia. Educational status of head of household, having radio, television and improved sanitation facility (latrine) at home and being in a better household wealth status were identified as a determinant for the presence of basic handwashing facilities in the community. It is mandatory to identify effective intervention measures and set strategies to increase community-level basic handwashing facilities. For instance, orienting the community on how to install basic handwashing facilities from locally available and affordable items through demonstration and sensitization campaigns is essential to increase the coverage. Persistent handwashing practice is beyond individual routines; it can be affected by the social and the wider environment. As a result, it is essential to understand barriers of handwashing including structural factors such as availability of the services and existing government policies, and psychosocial factors such as cultural and traditional norms.

## Acknowledgments

Authors are grateful to the Measure DHS, ICF International for providing the EDHS data for this analysis.

## Author Contributions

**Conceptualization:** Daniel Bogale Odo, Alemayehu Gonie Mekonnen.

**Data curation:** Daniel Bogale Odo.

**Formal analysis:** Daniel Bogale Odo.

**Investigation:** Daniel Bogale Odo.

**Methodology:** Daniel Bogale Odo, Alemayehu Gonie Mekonnen.

**Resources:** Daniel Bogale Odo, Alemayehu Gonie Mekonnen.

**Software:** Daniel Bogale Odo.

**Visualization:** Daniel Bogale Odo.

**Writing – original draft:** Daniel Bogale Odo.

**Writing – review & editing:** Daniel Bogale Odo, Alemayehu Gonie Mekonnen.

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
