## [Decision Letter · Decision Letter 0]

11 Aug 2020

PONE-D-20-19899

Availability and factors influencing community level handwashing facility in Ethiopia: An implication for prevention of infectious diseases

PLOS ONE

Dear Dr. Odo,

Thank you for submitting your manuscript to PLOS ONE. After careful consideration, we feel that it has merit but does not fully meet PLOS ONE’s publication criteria as it currently stands. Therefore, we invite you to submit a revised version of the manuscript that addresses the points raised during the review process.<>Please submit your revised manuscript by Sep 25 2020 11:59PM. If you will need more time than this to complete your revisions, please reply to this message or contact the journal office at plosone@plos.org. Please include the following items when submitting your revised manuscript:

We look forward to receiving your revised manuscript.

Kind regards,

Khin Thet Wai, MBBS, MPH, MA (Population & Family Planning Resear

Academic Editor

PLOS ONE

Journal Requirements:

2. Thank you for stating in the text of your manuscript "The survey was approved by the National Research Ethics Review Committee of Ethiopia and ORC Macro Institutional. Authors got permission from ICF-DHS program to use the DHS data and accessed through ‘" ext-link-type="uri" xlink:type="simple">https://www.dhsprogram.com/data/dataset_admin/login_main.cfm’." Please also add this information to your ethics statement in the online submission form.

In addition, in your ethics statement in the Methods section and in the online submission form, please provide additional information about the data used in your retrospective study. Specifically, please clarify whether all data from the DHS surveys were fully anonymized before you accessed them.

Also, please include the date(s) on which you accessed the database to obtain the data used in your study.

Reviewers' comments:

Reviewer's Responses to Questions

**Comments to the Author**

1. Is the manuscript technically sound, and do the data support the conclusions?

Reviewer #1: Yes

Reviewer #2: Yes

2. Has the statistical analysis been performed appropriately and rigorously? 

Reviewer #1: Yes

Reviewer #2: Yes

3. Have the authors made all data underlying the findings in their manuscript fully available?

Reviewer #1: Yes

Reviewer #2: Yes

4. Is the manuscript presented in an intelligible fashion and written in standard English?

Reviewer #1: Yes

Reviewer #2: Yes

5. Review Comments to the Author

Reviewer #1: Thank you for the opportunity to review this manuscript. Good paper, excellent study. This research is timely and fills a gap in the knowledge base and is worth of publication. Prior to publication, the article will require a careful editorial review for formatting and clarity, but most importantly the authors need to address potential bias and error that one must grapple with when dealing with large data sets. Specifically, very large data sets can lead to determine that a difference exists when it fact it does not.If there is a potential design flaw, such as missing IVs or other unknown characteristics of the sample, the effects of unknown contributors to the model are amplified as the data becomes larger. In the discussion, please consider limitations (in the limitations section) to the study and whether there are unknown characteristics that might potentially

Line 140, under Independent variable section, delete the extra “l” in Below.

Line 229 delete the space in “percent”. Also in line 306

Reviewer #2: 1.Title: I would recommend deletion of the word ‘an’ to read: Availability and factors influencing community level handwashing facility in Ethiopia: implication for prevention of infectious diseases

2.Introduction: I would recommend the use more recent literature in the introduction. I believe there more recent literature that can replace references 1, 4,5 and 6 [e.g. review by Willmott, M et al 2015]

3.Methods: Line 115 I would recommend deletion of the word “recent” and instead say… formed in 2007

4.Results:

i.The concepts of hot and cold spots used in line 173 page 10 need to be defined

ii.Table 3 appearing on page 10-12 may need to be reduced or formatted to fit in one page in its current format, it difficult to follow

iii.On page 10, line 182, the 54.81% of household heads with no formal education; it is not clear what the denominator is. Are we referring to the female headed household or the total sample

5.Discussion: The discussion section is well written, however:

i.This section would flow better if the authors started by summarizing the key results before starting the implication of the results

ii.I recommend inclusion of a brief discussion on relationship between availability of hand washing facility and household wealth which would allow more balance conclusions and recommendation. As it stands now the authors seem to suggest that improved awareness equals action/behavior (installation of hand washing facility). Although the authors rightfully acknowledge that awareness and knowledge alone are necessary but insufficient for changing behavior.

6. PLOS authors have the option to publish the peer review history of their article (what does this mean?). If published, this will include your full peer review and any attached files.

Reviewer #1: No

Reviewer #2: No

---

## [Author Response · Author response to Decision Letter 0]

25 Aug 2020

• We thought we have addressed all the concerns raised by our editor and reviewers. 

• The issues raised by the first reviewer are all incorporated in the document. 

• As it can be seen the submitted files, all editorial and formatting issues addressed.

---

## [Editor Report · Decision Letter 1]

28 Oct 2020

PONE-D-20-19899R1

Availability and factors influencing community level handwashing facility in Ethiopia: implication for prevention of infectious diseases

PLOS ONE

Dear Dr. Odo,

Thank you for submitting your manuscript to PLOS ONE. After careful consideration, we feel that it has merit but does not fully meet PLOS ONE’s publication criteria as it currently stands. Therefore, we invite you to submit a revised version of the manuscript that addresses the points raised during the review process.please by="" manuscript="" revised="" submit="" your=""/please

please by="" manuscript="" revised="" submit="" your=""

Please include the following items when submitting your revised manuscript:/please

We look forward to receiving your revised manuscript.

Kind regards,

Khin Thet Wai, MBBS, MPH, MA (Population Family Planning Resear

Academic Editor

PLOS ONE

Additional Editor Comments (if provided):

Authors have addressed the comments of reviewers comprehensively.

To strengthen scientific integrity and to uphold the publication standard of the journal, extensive English language editing is necessary.

---

## [Author Response · Author response to Decision Letter 1]

15 Nov 2020

Dear Khin Thet Wai,

We appreciate the opportunity to submit a revision. All the comments todate were helpful and we were able to address all of them. In the current version of our manuscript, we did extensive English language editing as per the comment from editor.

---

## [Editor Report · Decision Letter 2]

18 Nov 2020

Availability and factors influencing community level handwashing facility in Ethiopia: implication for prevention of infectious diseases

PONE-D-20-19899R2

Dear Dr. Odo,

We’re pleased to inform you that your manuscript has been judged scientifically suitable for publication and will be formally accepted for publication once it meets all outstanding technical requirements.

Kind regards,

Khin Thet Wai, MBBS, MPH, MA (Population Family Planning Resear

Academic Editor

PLOS ONE
---

## [Editor Report · Acceptance letter]

23 Nov 2020

PONE-D-20-19899R2 

Availability and factors influencing community level handwashing facility in Ethiopia: implication for prevention of infectious diseases 

Dear Dr. Odo:

I'm pleased to inform you that your manuscript has been deemed suitable for publication in PLOS ONE. Congratulations! Your manuscript is now with our production department. 

Kind regards, 

on behalf of

Dr. Khin Thet Wai 

Academic Editor

PLOS ONE